# Structural basis of p62/SQSTM1 helical filaments and their role in cellular cargo uptake

Arjen J. Jakobi [1,2,3,8], Stefan T. Huber [1,8,9], Simon A. Mortensen[1,4,5,9], Sebastian W. Schultz[6], Anthimi Palara[7], Tanja Kuhm[1,8], Birendra Kumar Shrestha[7], Trond Lamark [7], Wim J.H. Hagen[1], Matthias Wilmanns [2,3], Terje Johansen [7], Andreas Brech[6] & Carsten Sachse [1,4,5]*

p62/SQSTM1 is an autophagy receptor and signaling adaptor with an N-terminal PB1 domain that forms the scaffold of phase-separated p62 bodies in the cell. The molecular determinants that govern PB1 domain filament formation in vitro remain to be determined and the role of p62 filaments inside the cell is currently unclear. We here determine four high-resolution cryo-EM structures of different human and Arabidopsis PB1 domain assemblies and observed a filamentous ultrastructure of p62/SQSTM1 bodies using correlative cellular EM. We show that oligomerization or polymerization, driven by a double arginine finger in the PB1 domain, is a general requirement for lysosomal targeting of p62. Furthermore, the filamentous assembly state of p62 is required for autophagosomal processing of the p62-specific cargo KEAP1. Our results show that using such mechanisms, p62 filaments can be critical for cargo uptake in autophagy and are an integral part of phase-separated p62 bodies.

[1] European Molecular Biology Laboratory (EMBL), Structural and Computational Biology Unit, Meyerhofstraße 1, 69117 Heidelberg, Germany. [2] European Molecular Biology Laboratory (EMBL), Hamburg Unit c/o DESY, Notkestraße 85, 22607 Hamburg, Germany. [3] The Hamburg Centre for Ultrafast Imaging (CUI), Luruper Chaussee 149, 22761 Hamburg, Germany. [4] Ernst-Ruska Centre for Microscopy and Spectroscopy with Electrons (ER-C-3/Structural Biology), Forschungszentrum Jülich, 52425 Jülich, Germany. [5] JuStruct: Jülich Center for Structural Biology, Forschungszentrum Jülich, 52425 Jülich, Germany. [6] Department of Molecular Cell Biology, Institute for Cancer Research, Oslo University Hospital, Montebello, N-0379 Oslo, Norway. [7] Molecular Cancer Research Group, Institute of Medical Biology, University of Tromsø – The Arctic University of Norway, 9037 Tromsø, Norway. [8]Present address: Department of Bionanoscience, Kavli Institute of Nanoscience, Delft University of Technology, Van der Maasweg 9, 2629 HZ Delft, The Netherlands. [9]These authors contributed equally: Stefan T. Huber, Simon A. Mortensen. *email: c.sachse@fz-juelich.de

p62/SQSTM1 (from hereon p62) is a multifunctional adaptor protein that acts as a central scaffold protein in different cellular processes, such as autophagy and signaling[1]. p62 has a tendency to cluster, and in human cells, is often observed in discrete punctae known as p62 bodies[2]. The formation of these bodies is dependent on the amino-terminal PB1 domain of p62[2]. PB1 domains are protein interaction modules with critical roles in the assembly of protein complexes involved in autophagy, signaling, cell division, and redox processes[3], as well as the auxin-response pathway in plants[4]. PB1 domains form homotypic interactions via conserved electrostatic motifs molded by basic or acidic surface patches on opposite faces of their ubiquitin-like β-grasp fold[2,5]. According to their interaction profile, PB1 domains are classified into type A (acidic, OPCA motif), type B (basic), or mixed-type AB members[5]. While type A and type B PB1 domains can form heterodimeric protein complexes, type AB members can mediate interactions with either PB1 domain type or engage in homotypic interactions to form homo-oligomers or hetero-oligomers[2,6]. More recently, PB1-mediated self-interaction of p62/SQSTM1 was found to result in the formation of filamentous polymers[7] with helical symmetry in vitro[8].

p62 has been shown to function in autophagy and cellular signaling. Autophagy is a degradative cellular housekeeping pathway by which cytoplasmic materials are engulfed in a double-membrane vesicle termed the autophagosome and delivered to the lysosomal compartment[9]. Substrates for autophagy are not limited by molecular size and include large protein aggregates, intracellular pathogens, and cellular organelles. Selective autophagy has been characterized as the process that specifically directs cytosolic substrates to the formation site of autophagosomal membranes[10,11]. As an autophagy receptor, p62 links cargo proteins with the autophagosome membrane. PB1-mediated oligomerization of p62 is essential for its function as a selective autophagy receptor[12] and thought to facilitate co-aggregation of ubiquitylated cargo[13]. The C-terminal UBA domain of p62 captures ubiquitinated cargo, and the LIR motif guides the cargo–receptor complex to Atg8/LC3, which is anchored to the surface of the autophagosomal membrane[14,15]. Importantly, in addition to the selective autophagy degradation of ubiquitinated cargo, p62 is also involved in the degradation of other substrates such as KEAP1 known as a regulator of the antioxidative stress response transcription factor NRF2. KEAP1 binds directly to a specific motif in p62, i.e., the KEAP1-interacting region (KIR)[16,17]. In signaling, p62 bodies constitute an interaction hub for the kinases MEKK3, MEK5, and aPKCs, which also contain PB1 domains[2], in addition to triggering the NF-κB pathway through the polyubiquitination of tumor necrosis factor (TNF) receptor-associated factor 6 (TRAF6)[18].

Due to p62's involvement in protein homeostasis, the impairment of autophagy or oxidative stress results in aggregation or upregulation of p62, including increased body formation[19,20]. Recently, we and others independently found that p62 reconstituted with other components of the autophagy pathway, such as ubiquitinated model cargo, and the selective autophagy receptor NBR1, spontaneously coalesces into p62 bodies in vitro[21] and shows the characteristics of liquid–liquid-phase separation in vivo[22]. These studies established that oligomerization by the N-terminal PB1 domain of p62 is an essential requirement for recapitulating phase separation in vitro, as well as for cargo uptake in vivo[12,22].

The exact structural requirements and physiological conditions under which p62-PB1 domains self-assemble or engage in hetero-PB1 complexes are currently unclear. Furthermore, it is not known what assembly state of p62 is required for biological functions such as cargo uptake in autophagy or the formation of phase-separated compartments in vivo. Based on high-resolution electron cryo-microscopy (cryo-EM) and crystal structures, cellular EM, biochemical, and cellular characterization, we here revealed the structural basis for polymeric PB1 self-assembly and defined the relevance of symmetry and spatial arrangement of the polymeric assembly state for p62 autophagy function in vivo.

## Results

**p62, TFG1, and AtNBR1–PB1 domains form filamentous polymers.** Based on our previous finding that p62 is capable of forming homo-oligomeric filamentous assemblies[8], we set out to understand whether related AB-type PB1 domains possess a similar property to self-assemble. With reference to sequence alignments (Fig. 1a), we expressed and purified PB1 domains from human p62$^{1-102}$, p62$^{1-122}$, TFG1$^{1-95}$ (Trk-fused gene 1), the atypical protein kinase PKCζ$^{11-101}$, as well as the evolutionary-related PB1 domain of the NBR1$^{1-94}$ autophagy receptor from *Arabidopsis thaliana* (AtNBR1)[23]. p62, TFG1, PKCζ, and AtNBR1 are multi-domain proteins that share the N-terminal PB1 domain with additional functional C-terminal domains (Fig. 1b). In order to assess whether these PB1 domain-containing proteins are capable of forming high-molecular-weight assemblies, we performed sedimentation assays by ultracentrifugation. The PB1 domains of TFG1$^{1-95}$, AtNBR1$^{1-94}$, p62$^{1-102}$, and p62$^{1-122}$ were found in the pellet fraction, whereas PB1 domains from PKCζ remained soluble (Fig. 1c), which is in agreement with our previous study, showing that both p62$^{1-102}$ and p62$^{1-122}$ form filamentous structures[8]. Furthermore, we visualized the pelleted fractions by using negative staining electron microscopy (EM) and observed elongated filamentous or tubular assemblies for the PB1 domains of p62$^{1-122}$, TFG1, and AtNBR1 that measure 145 ± 5, 900 ± 52, and 120 ± 4 Å in diameter, respectively (Fig. 1d). Closer inspection of the sequence alignments revealed that all three of these PB1 domains share the tandem arginine motif close to the canonical lysine residue of the basic motif in B-type PB1 domains. By contrast, this tandem arginine motif is absent in AB-type PB1 sequences of PKCζ that does not form filamentous or tubular structures, suggesting a critical role for self-assembly.

**Cryo-EM structures of AtNBR1 and p62-PB1 filaments.** Of the three PB1 assemblies studied, AtNBR1$^{1-94}$ (AtNBR1–PB1) and p62$^{1-122}$ (p62-PB1) formed homogeneous filaments of constant diameter that appeared best suited for high-resolution structure investigation by cryo-EM. Therefore, we vitrified filaments of purified AtNBR1–PB1 and p62-PB1 domains and imaged the samples by cryo-EM (Fig. 2a, b). Image classification of segmented PB1 helices revealed that both AtNBR1–PB1 and p62-PB1 polymerize in two different tubular morphologies: a projection class with a ladder-like pattern, we term L-type, and a projection class with a serpent-like one, we term S-type (Fig. 2c; Supplementary Fig. 1A–C). L-type and S-type helices partition approximately evenly, i.e., 40–60% and 55–45% for p62-PB1 and AtNBR1–PB1 samples, respectively. Further analysis revealed that the occurrence of L-type or S-type assemblies is persistent along the individual helices in micrographs of AtNBR1–PB1, whereas for p62-PB1 filaments regularly displayed transitions from L-type to S-type symmetry (Supplementary Fig. 1D). In an effort to understand the underlying structures of L-type and S-type projections, we analyzed the averaged power spectra from in-plane rotated segments and from class averages. The best Fourier spectra of AtNBR1–PB1 and p62-PB1 showed discrete layer-line reflections up to 5.9 and 4.7 Å, suggesting a helical organization and preservation of structural order up to high resolution (Supplementary Fig. 1E, F). The comparison of the Fourier spectra

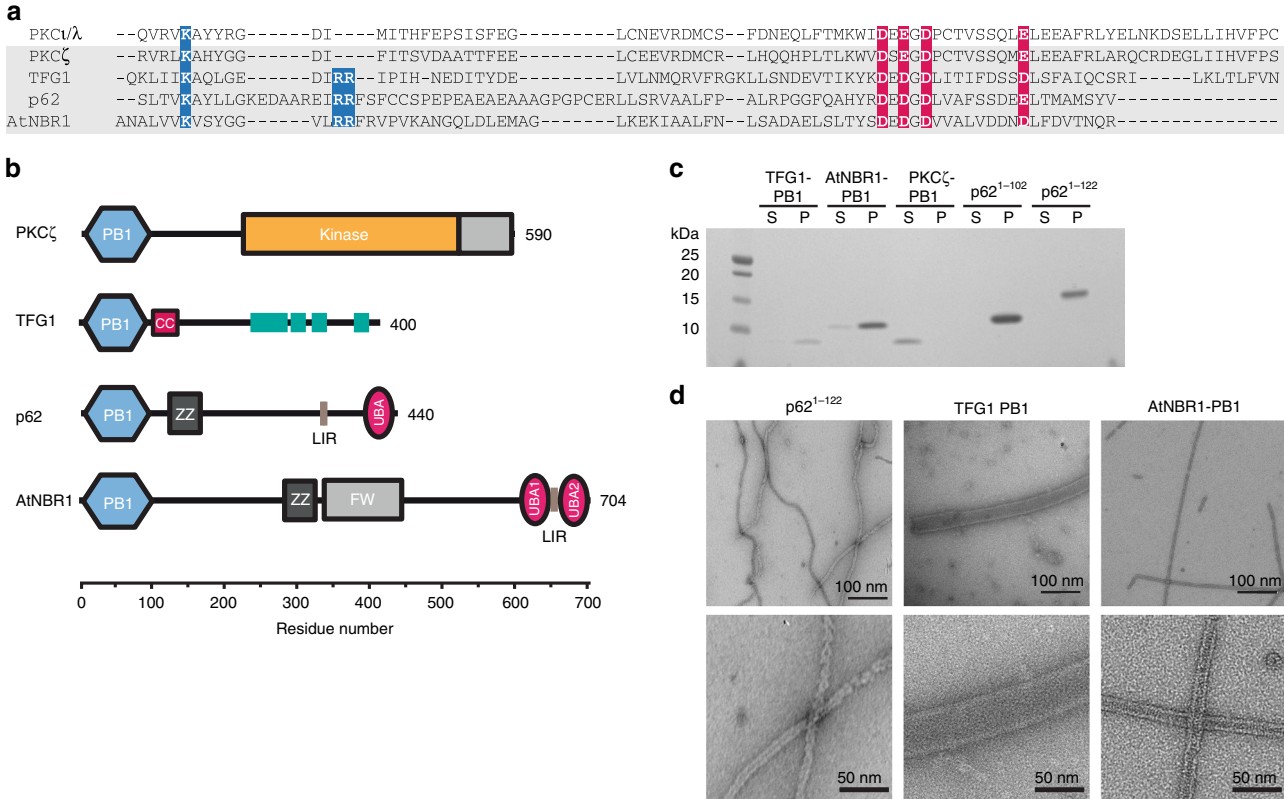

**Fig. 1 Type A/B PB1 domains and their capability to form polymers. a** Sequence alignment of the type A/B PB1 domains with highlighted tandem arginine motif (blue) in addition to basic (blue) and acidic residues (red). **b** Domain architecture of PKCz, TFG1, p62, and AtNBR1 proteins. **c** Pelletation assay of purified type A, B, or AB PB1 domains: TFG1, AtNBR1, PKCζ, p62$^{1-102}$, and p62$^{1-122}$. Corresponding lanes of soluble (S) and pellet (P) fraction are shown. Only PKCζ remains soluble, whereas TFG1, AtNBR1, and p62 are found in the pellet. Source data are provided as a Source Data file. **d** Electron micrographs of negatively stained specimens reveal elongated filamentous p62$^{1-122}$, tubular polymers of TFG1 and AtNBR1 of 145 ± 5, 900 ± 52, and 120 ± 4 Å nm in diameter, respectively.

confirmed that L-type and S-type structures are differently organized in their helical lattice. By indexing the layer lines in the Fourier spectra of AtNBR1–PB1 filaments, we concluded that L-type is a two-stranded helix with a pitch of 77.2 Å and 11.47 subunits/turn, whereas S-type is a single double-stranded helix with a pitch of 68.2 Å and 11.55 subunits/turn. For p62-PB1, we observed a four-stranded L-type assembly and a three-stranded S-type assembly. In the latter S-type, one of the three helical rungs is propagating in an antiparallel orientation, related to the central rung by local dihedral symmetry. The L-type here has a pitch of 135.9 Å with 14.16 subunits/turn, and S-type has a pitch of 138.6 Å with 13.60 subunits/turn. Using the derived symmetries, we determined the 3.5/3.9- (L-type, p62/AtNBR1) and 4.0/4.4 Å- (S-type, p62/AtNBR1) resolution structures (Fig. 2c, Tables 1, 2; Supplementary Fig. 1G, H). All four structures form tubules of ~120 Å and 150-Å width with an inner diameter of 45 Å and 70 Å for AtNBR1–PB1 and p62-PB1, respectively. In all reconstructions, the main chain of the PB1 domain could be resolved with α-helical pitch features and individual β-strands separated. The overall fold of the asymmetric unit was found compatible with the NMR structure of the p62-PB1 monomer[24,25] (Fig. 3a, b). In the absence of prior structural information, we traced the AtNBR1–PB1 de novo. This de novo-built model is in close agreement with the 1.6-Å crystal structure of a polymerization-deficient AtNBR1–PB1 mutant, which we solved in parallel (Table 3; Supplementary Fig. 2A). The relative orientation between adjacent subunits is very similar in the respective S-type and L-type assemblies of AtNBR1–PB1 and p62-PB1 (Supplementary Fig. 2B). The β1–α1 loop in p62 is

flexible and only visible in the L-type assembly density (Supplementary Fig. 2C). Expanding the asymmetric unit by using the helical parameters of the L-type and S-type structures allowed analysis of the interface between repeating units. Despite overall similar interaction modes, the AtNBR1 and p62 assemblies showed differences in relative domain rotation between adjacent subunits and with respect to the helical axis (Fig. 3c). In agreement with sequence analysis (see Fig. 1a), the electrostatic potential mapped onto the molecular surface of the structures revealed that opposing charged surfaces mediate the PB1–PB1 interactions in the helical repeat (Fig. 3d). In addition, we more closely examined the interface of homomeric interactions in the helical assemblies. The main interactions are formed between a double arginine finger formed by two neighboring arginine residues in strand β2 (R19–R20$^{AtNBR1}$/R21–22$^{p62}$) stabilizing strong salt bridges to acidic residues (D60/D62/D64/D73$^{AtNBR1}$ or D69/D71/D73/E82$^{p62}$) in the OPCA motif located in the β2–β3 loop and the α2 helix (Fig. 3e). These interactions are assisted by the canonical-type B lysine (K11$^{AtNBR1}$ and K7$^{p62}$) in strand β1. Free-energy calculations using the PDBePISA server[26] suggest that a large part of the interface free energy is contributed by the double arginine finger. In addition to the canonical transverse interactions, the helices are further stabilized by longitudinal interactions Y14$^{AtNBR1}$/N28$^{AtNBR1}$ or K102$^{p62}$/D92$^{p62}$ and R59$^{p62}$/D93$^{p62}$ to subunits of neighboring strands along the helical axis (Supplementary Fig. 2D, E). The importance of electrostatic interactions on filament stability is further supported by the observation that increased ionic strength impedes stable filament formation and is sensitive to pH (Supplementary

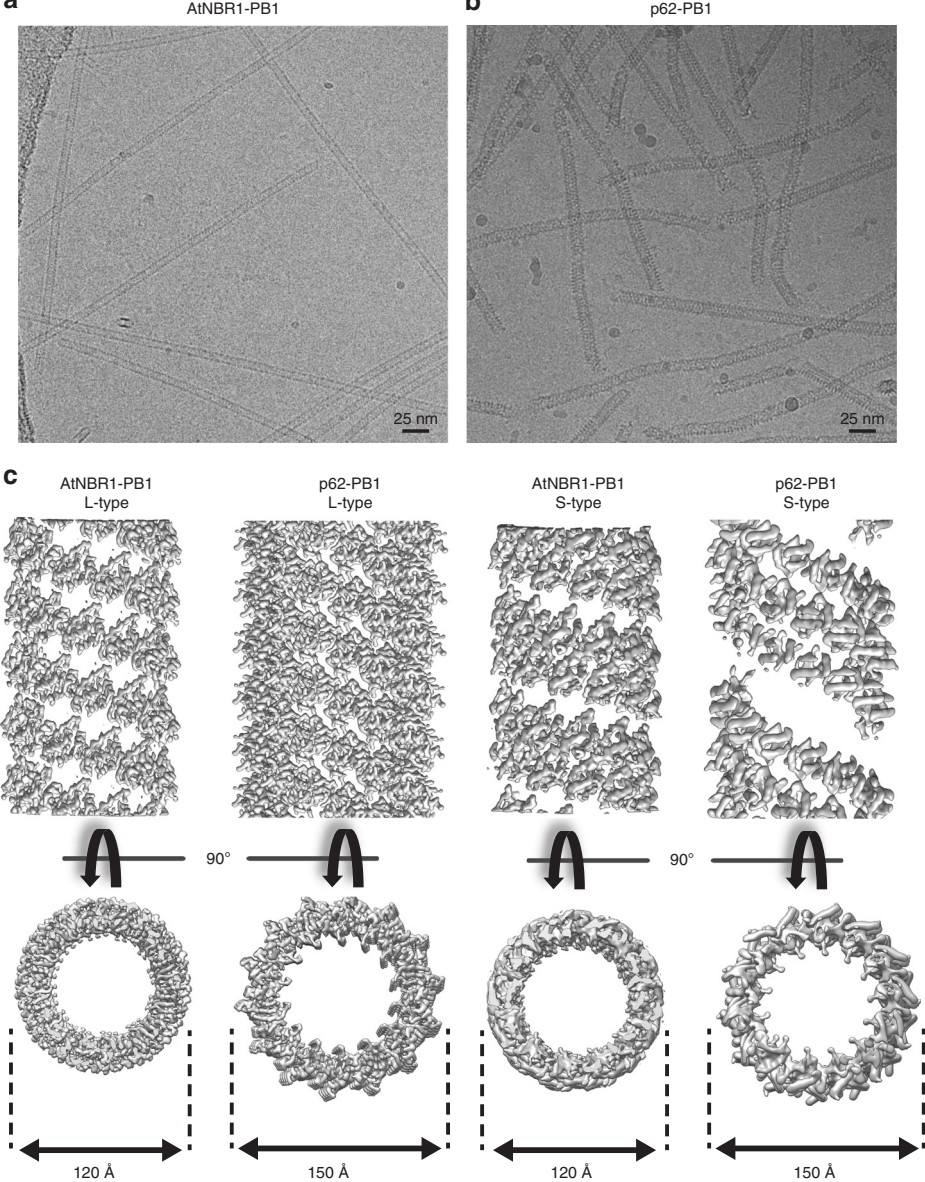

**Fig. 2 Cryo-EM structures of AtNBR1[11–94] and p62[1–122].** **a** Electron cryo-micrograph of AtNBR1–PB1[1–94] and (**b**) p62-PB1[1–122] assemblies. **c** Side and top views for determined cryo-EM structures of L-type AtNBR1–PB1 (far left), p62-PB1 (left), and S-type assembly of AtNbr–PB1 (right), p62-PB1 (far right).

Fig. 3A–H). To validate our structural interpretation, we performed pull-down experiments using MBP-tagged wild-type AtNBR1–PB1 as a prey and a series of AtNBR1–PB1 interface mutants as bait (Fig. 3f). All interface mutants decrease binding significantly compared with the wild type, and binding is completely abrogated in mutants lacking the double arginine finger, in agreement with observations in cellular assays[2,23]. Together, the cryo-EM structures of two PB1 domain assemblies reveal that in addition to the canonical-type electrostatic AB interactions, the self-polymerization property is linked to the presence of a double arginine finger.

**PB1 domain interactions in the context of filamentous p62.** After establishing the molecular basis of PB1 domain homopolymerization, we wanted to understand how these assemblies interact with other PB1 domains of the A and B types that have been shown to co-localize with p62 punctae[2]. We therefore expressed and purified A-type human PB1 domains of

MEK5[5–108] and NBR1[1–85], the B-type PB1 domain of MEKK3[43–127], and the AB-type PB1 domain of PKCζ[11–101] and determined their binding affinities for polymerization-deficient p62[1–102] (D69A/D73A)[5] by isothermal titration calorimetry (ITC). These PB1 domains show 2–10-fold lower binding affinity to p62 compared with its self-interaction dissociation constant ($K_D$) of 6 nM[27], with $K_D$ of $8.9 \pm 0.9$ nM, $12.6 \pm 0.4$ nM, $26.8 \pm 0.5$ nM, and $105 \pm 1.3$ nM determined for PKCζ[27], NBR1, MEKK3, and MEK5, respectively (Fig. 4a). Moreover, other PB1 interactions, such as binding of NBR1–PB1 to MEKK3, have also been measured and have even lower affinity ($K_D$ of 13.3 μM). We therefore hypothesized that binding of p62-interacting PB1 domains could compete with p62 self-polymerization and affect the assembly structures of p62-PB1 filaments. We found that NBR1–PB1 strongly interacts with p62-PB1 filaments and shortens p62-PB1 filaments on average to less than half the starting length (Fig. 4b, c). Surprisingly, MEKK3, MEK5, and PKCζ–PB1 showed no effect on the pelletation behavior of p62 assemblies, although having only marginally lower affinities than

NBR1 (Fig. 4d). To further analyze the interactions, we turned to negative staining EM. In agreement with the co-sedimentation data, for PB1 domains other than NBR1 we did not observe any effect on the morphology of p62-PB1 filaments and the measured filament lengths. In order to increase the sensitivity of detecting interactions with p62-PB1 filaments, we also imaged p62-PB1 filaments incubated with nanogold-labeled NBR1, MEKK3, MEK5, and PKCζ PB1 domains using negative staining EM (Fig. 4e). For all PB1 domains, the micrographs confirmed end-on binding of the PB1 domains to p62-PB1 polymers or to oligomeric, ring-like structures. Interestingly, NBR1, MEK5, and PKCζ PB1 domains preferably bind to one end of the filament (Fig. 4f), consistent with an overall polar assembly observed in the 3D reconstructions of p62-PB1 filaments (see Fig. 2). MEKK3–PB1 (type B) was not observed at p62-PB1 filament ends, but occasionally found at oligomeric ring-like structures. Biochemical interaction studies suggest that assembled filamentous p62 can display significantly lower apparent binding affinities for interacting PB1 domains than when present in the monomeric form.

**Cellular p62 bodies consist of filamentous structures**. Although self-oligomerization of p62 has been shown to be essential for targeting of p62 to the autophagosome[12], it is unclear whether the filamentous assemblies observed in vitro are involved in this process or even occur inside of cells. We used correlative light and electron microscopy (CLEM) to study the ultrastructure of p62 bodies in a targeted manner. In order to enrich endogenous p62 bodies in RPE1 cells, we overexpressed a human NBR1-D50R mutant that abolishes the interaction with p62[2]. Co-sedimentation experiments, in which the relative amount of p62 in the monomeric and polymeric state are determined, indeed showed that wild-type NBR1 solubilizes filamentous p62-PB1, whereas the D50R mutant does not (Fig. 5a). In RPE1 cells, the NBR1-D50R mutant consistently produced larger p62 clusters possibly by promoting self-polymerization as observed in vitro (Supplementary Fig. 4A). In such cells, we localized p62 to punctate areas of $0.5 \pm 0.1$-μm diameter by fluorescence microscopy and visualized their ultrastructure by electron tomography (Fig. 5b, Supplementary Fig. 4B, C). The electron micrographs revealed that p62 bodies have a distinct appearance that is well differentiable from the cytosol with an electron-dense boundary of ~60-nm thickness surrounding the body (Fig. 5c, d). We thresholded the interior density and found that the p62 bodies are composed of a dense meshwork of filamentous assemblies (Fig. 5e). Quantitative analysis of thresholded images confirmed the presence of elongated filament-like structures with an average diameter of 15 nm compatible in dimensions with the helical p62 structures observed in vitro[8]. We estimated the length of these structures by tracing individual filaments in

**Table 1 Cryo-EM data collection and helical reconstruction.**

| | AtNBR1–PB1$^{1-94}$ (S-type: EMD-10500, L-type: EMD-10499) | p62-PB1$^{1-122}$ (S-type: EMD-10502, L-type: EMD-10501) |
|---|---|---|
| *Data collection and processing* | | |
| Magnification | 105kx | 130kx |
| Voltage (kV) | 300 | 300 |
| Electron exposure (e⁻/Å) | 17 | 40 |
| Defocus range (μm) | 1.0–4.0 | 0.5–2.5 |
| Pixel size (Å) | 1.386 | 1.040 |
| Symmetry imposed | S-type: C1 L-type: C2 | S-type: C1 L-type: C2 |
| Final no. of segments | S-type: 18,021 L-type: 25,387 | S-type: 51,679 L-type: 51,853 |
| Helical rise (Å) | S-type: 5.905 L-type: 6.721 | S-type: 9.78 L-type: 4.787 (9.574)* |
| Helical twist (˚) | S-type: −31.17 L-type: −31.44 | S-type: −26.48 L-type: 77.29 (−25.42)* |
| Global map resolution (Å, FSC = 0.143) | S-type: 4.4 L-type: 3.9 | S-type: 4.0 L-type: 3.5 |
| Local map resolution range (Å) | S-type: 4.0–4.7 L-type: 3.4–4.1 | S-type: 3.7–4.4 L-type: 3.3–4.4 |

*Equivalent notation for asymmetric unit of two monomers as described in the main text

**Table 2 Model refinement statistics.**

| | AtNBR1–PB1$^{1-94}$ (S-type: PDB ID 6TGP, L-type: PDB ID 6TGN) | p62-PB1$^{1-122}$ (S-type: PDB ID 6TH3, L-type: PDB IC 6TGY) |
|---|---|---|
| *Model refinement* | | |
| Initial model used (PDB code) | PDB-6TGS (X-ray model) | PDB ID 2KKC# |
| Model resolution (Å, FSC = 0.5) | S-type: 5.5 L-type: 4.3 | S-type: 4.0 L-type: 3.6 |
| Map-sharpening B-factor (Å²) | S-type: −300 L-type: −200 | S-type: −193 L-type: −139 |
| *Model composition* | | |
| Non-hydrogen atoms | 669 (S-/L-type) | 808 (S-/L-type) |
| Protein residues | 88 (S-/L-type) | 104 (S-/L-type) |
| *R.m.s. deviations* | | |
| Bond lengths (Å) | 0.009/0.008 (S-/L-type) | 0.006/0.007 (S-/L-type) |
| Bond angles (˚) | 1.16/1.161 (S-/L-type) | 1.22/1.24 (S-/L-type) |
| *Validation* | | |
| MolProbity score | 2.41/2.29 (S-/L-type) | 1.94/1.64 (S-/L-type) |
| Clashscore* | 7.59/6.41 (S-/L-type) | 4.89/1.88/ (S-/L-type) |
| Rotamer outliers (%) | 1.41/1.41 (S-L-type) | 0.00/0.63 (S-/L-type) |
| Ramachandran plot | | |
| Favored (%) | 93.21/94.19 (S-/L-type) | 83.33/83.33 (S-/L-type) |
| Allowed (%) | 6.79/5.81 (S-/L-type) | 16.67/16.67 (S-/L-type) |
| Disallowed (%) | 0.00 (S-/L-type) | 0.00 (S-/L-type) |

*Computed for 9-mer
#Saio et al.[24]

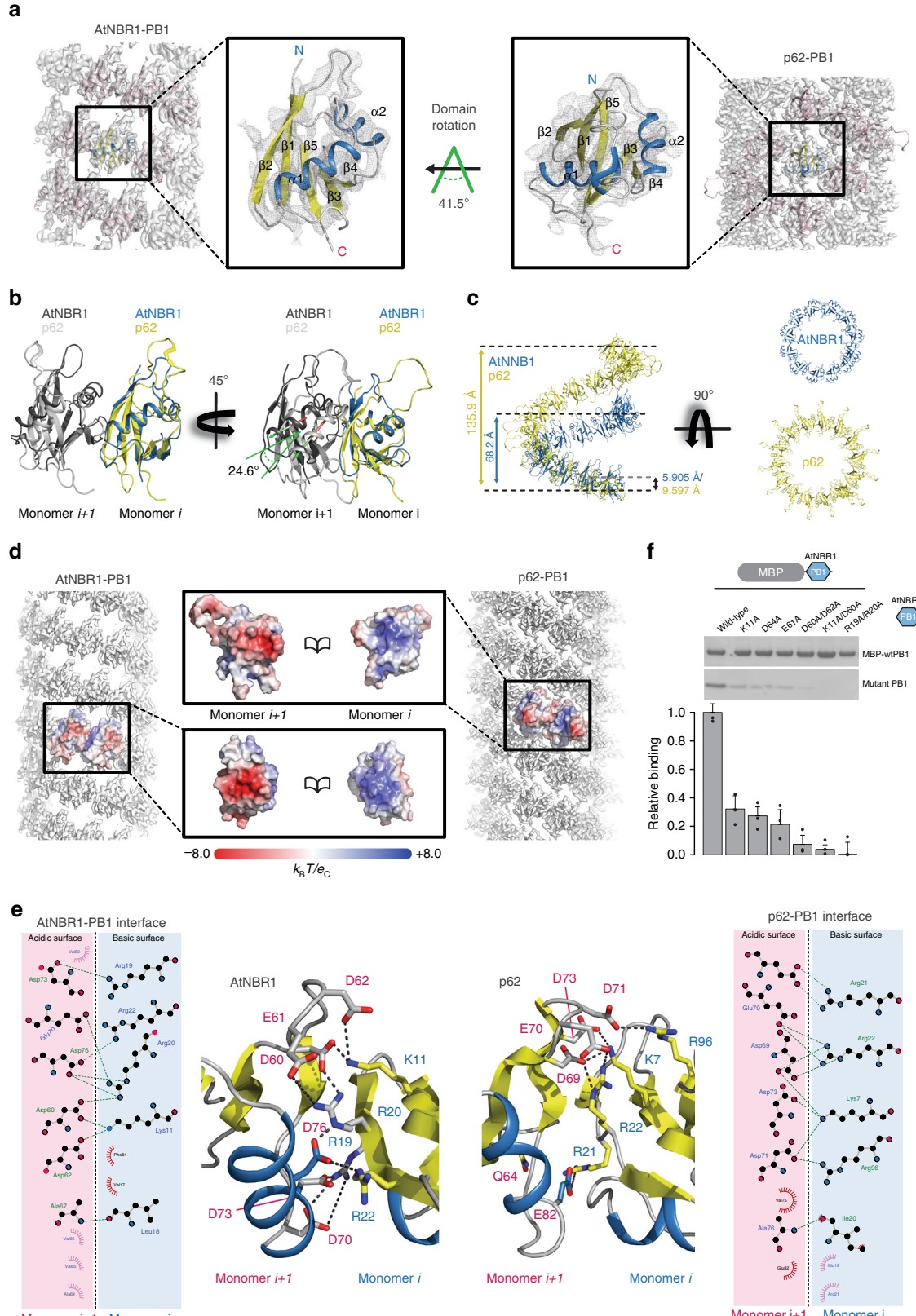

sequential tomogram slices (Fig. 5f). CLEM visualization of p62 bodies in cells under endogenous p62 levels confirms the presence of filamentous assemblies.

**The effect of different p62 assemblies on autophagy clearance.** We next set out to assess the relevance of symmetry and assembly

state of PB1-mediated filaments for biological function within cellular p62 bodies and lysosomal targeting through the autophagy pathway. In the comparison of PB1 assemblies visualized by negative staining EM, TFG1 showed the most striking difference to p62 assemblies both in size and apparent symmetry (see Fig. 1d). Therefore, we reasoned that a p62 chimera, in which

**Fig. 3 Structural basis of PB1 polymer formation. a** Cryo-EM structures of AtNBR1–PB1 (left) and p62-PB1 filaments are shown with atomic ribbon models (α-helix: blue and β-strands: yellow) superposed on the density. Close-ups show that both PB1 domains display the canonical ubiquitin-like fold (center left and center right). The arrow indicates the rotation of the p62-PB1 subunit relative to the AtNBR1–PB1 subunit in their respective assemblies. **b, c** Differences in the PB1–PB1 interface give rise to different helical architectures. (Left) Monomer *i* of AtNBR1 (blue) and monomer *i* of p62 (yellow) were superposed to visualize the degree of domain rotation toward the next monomer along the helical rung (monomer i + 1). (Right) Adjacent subunits along the helical rung for AtNBR1 display a 25° inward rotation compared with adjacent subunits of p62, explaining the observed differences in helical symmetry and diameter of AtNBR1–PB1 and p62-PB1 filaments, respectively (**c**). **d** Electrostatic potential surface of the determined AtNBR1–PB1 and p62-PB1 structures. For both structures, the propagation of the helical structure is mediated and stabilized by positively (blue) and negatively charged (red) surfaces on opposite faces of the PB1 fold. **e** Schematic illustration and detailed interactions of the PB1–PB1 interface as determined from the AtNBR1–PB1 and p62-PB1 cryo-EM structures, respectively. The structures are shown in cartoon representation highlighting key electrostatic residue contacts shown as sticks. **f** In vitro pulldown with maltose-binding protein (MBP)-tagged wild-type AtNBR1–PB1 of structure-based AtNBR1–PB1 domain mutants. Error bars represent standard deviation (SD) of three independent experiments. Source data are provided as a Source Data file.

### Table 3 X-ray crystallography data collection and refinement statistics.

| | |
|---|---|
| *Data collection statistics* | |
| Wavelength | |
| Resolution range | 37.9–1.53 (1.59–1.53) |
| Space group | P 21 21 2 |
| Unit cell | 43.13 79.44 24.14 90 90 90 |
| Total reflections | 25,830 (2499) |
| Unique reflections | 13,035 (1271) |
| Multiplicity | 2.0 (2.0) |
| Completeness (%) | 99.22 (99.30) |
| Mean I/sigma(I) | 10.45 (1.42) |
| Wilson B-factor | 20.99 |
| R-merge | 0.02799 (0.4132) |
| R-meas | 0.03958 (0.5844) |
| R-pim | 0.02799 (0.4132) |
| CC1/2 | 0.999 (0.655) |
| CC* | 1.00 (0.89) |
| *Model refinement* | |
| Reflections used in refinement | 13,030 (1271) |
| Reflections used for R-free | 669 (53) |
| R-work | 0.2456 (0.3525) |
| R-free | 0.2776 (0.4166) |
| CC (work) | 0.927 (0.750) |
| CC (free) | 0.902 (0.462) |
| *Model refinement* | |
| Number of non-hydrogen atoms | 816 |
| Macromolecules | 723 |
| Ligands | 52 |
| Solvent | 41 |
| Protein residues | 88 |
| RMS (bonds) | 0.007 |
| RMS (angles) | 0.79 |
| Ramachandran | |
| Favored (%) | 100.00 |
| Allowed (%) | 0.00 |
| Outliers (%) | 0.00 |
| Rotamer outliers (%) | 6.49 |
| Clashscore | 2.97 |
| Average B-factor | 32.42 |
| Macromolecules | 30.25 |
| Ligands | 51.32 |
| Solvent | 46.80 |

*Statistics for the highest-resolution shell are shown in parentheses

we exchange the native PB1 domain for TFG1–PB1, could clarify the role of the helical PB1 scaffold in autophagy clearance. We generated two p62 chimeras by fusing the TFG1–PB1 domain to either p62 (123–408) or p62Δ123–319 (mini-p62), containing only the p62 LIR motif and UBA domain (Fig. 6a) and visualized the resulting assemblies by negative staining EM (Fig. 6b). The TFG1:p62 chimera forms 48-nm wide filaments, which is approximately three times the diameter of WT-p62 filaments and possesses a helical architecture clearly different from that of WT-p62 filaments. The TFG1-mini-p62 chimera forms defined, ring-shaped oligomers with ~12 nm in diameter. To test whether the TFG1-p62 fusion constructs are able to form p62 bodies in cells, we expressed the chimeras fused to an N-terminal GFP tag in HeLa cells deficient of endogenous p62. As controls, we also expressed GFP-tagged WT-p62 and the mini-p62 construct (p62Δ123–319) (Fig. 6c). The transfected cells were analyzed by confocal fluorescence microscopy 24 h and 48 h post transfection. All constructs formed p62 bodies, with the majority of dots having a diameter in the range of 0.1–0.5 μm. We further classified GFP-positive punctae according to frequency of occurrence, the tendency to cluster, and the morphological appearance (Fig. 6c, d; Supplementary Fig. 5A).

We next asked whether TFG1-p62 could perform the biological function of p62. We first assessed whether TFG1-p62 can be turned over by autophagy and targeted to acidified cellular compartments by using the "traffic light" reporter. Here, the mCherry-YFP tandem tag is fused to the target protein, and the acidification of the construct in lysosomes is monitored by appearance of red punctae. Although both TFG1-p62 chimeras displayed a diffuse yellow fraction, they were almost as efficiently degraded by autophagy as the WT and mini-p62 constructs (Fig. 6e, f; Supplementary Fig. 5B–F). We then asked if the TFG1-p62 chimera was able to act as a cargo receptor for a p62-specific substrate, KEAP1, and mediate autophagy degradation. KEAP1 was shown to be entirely diffusely localized when expressed in cells lacking p62[16]. We first verified that purified KEAP1–DC domain still binds to the p62–TFG1 chimeras by using a pull-down assay (Supplementary Fig. 5G). Next, we monitored co-localization in cells and found that in analogy to biochemical binding data, the TFG1-p62 chimera, WT, and mini-p62 constructs co-aggregated with KEAP1 in cells, but only the WT and mini-p62 constructs could mediate acidification of tandem tagged KEAP1 when co-expressed as Myc-tagged constructs in the p62 KO HeLa cells. At the same time, no autophagic turnover of mCherry-YFP-KEAP1, however, was observed upon co-expression with chimera Myc-TFG1-p62 or Myc-TFG1-mini-p62 (Fig. 6g, h; Supplementary Fig. 6A–D). When we compromised the formation of PB1 domain-mediated filament assemblies by mutating p62's double arginine finger (R21A/R22A), p62 was completely diffusely localized and not degraded by autophagy (Supplementary Fig. 7A, B, Supplementary Fig. 8, Supplementary Movies 1, 2). This mutant also failed to mediate aggregation and autophagic degradation of KEAP1 in co-transfected cells (Supplementary Fig. 7C). In conclusion, although TFG1-p62 chimera can be degraded by autophagy despite their assembly into nonnative polymers, these assemblies are evidently unable to mediate degradation of the p62-specific substrate KEAP1 in analogy to the polymerization-deficient double arginine finger mutant of p62.

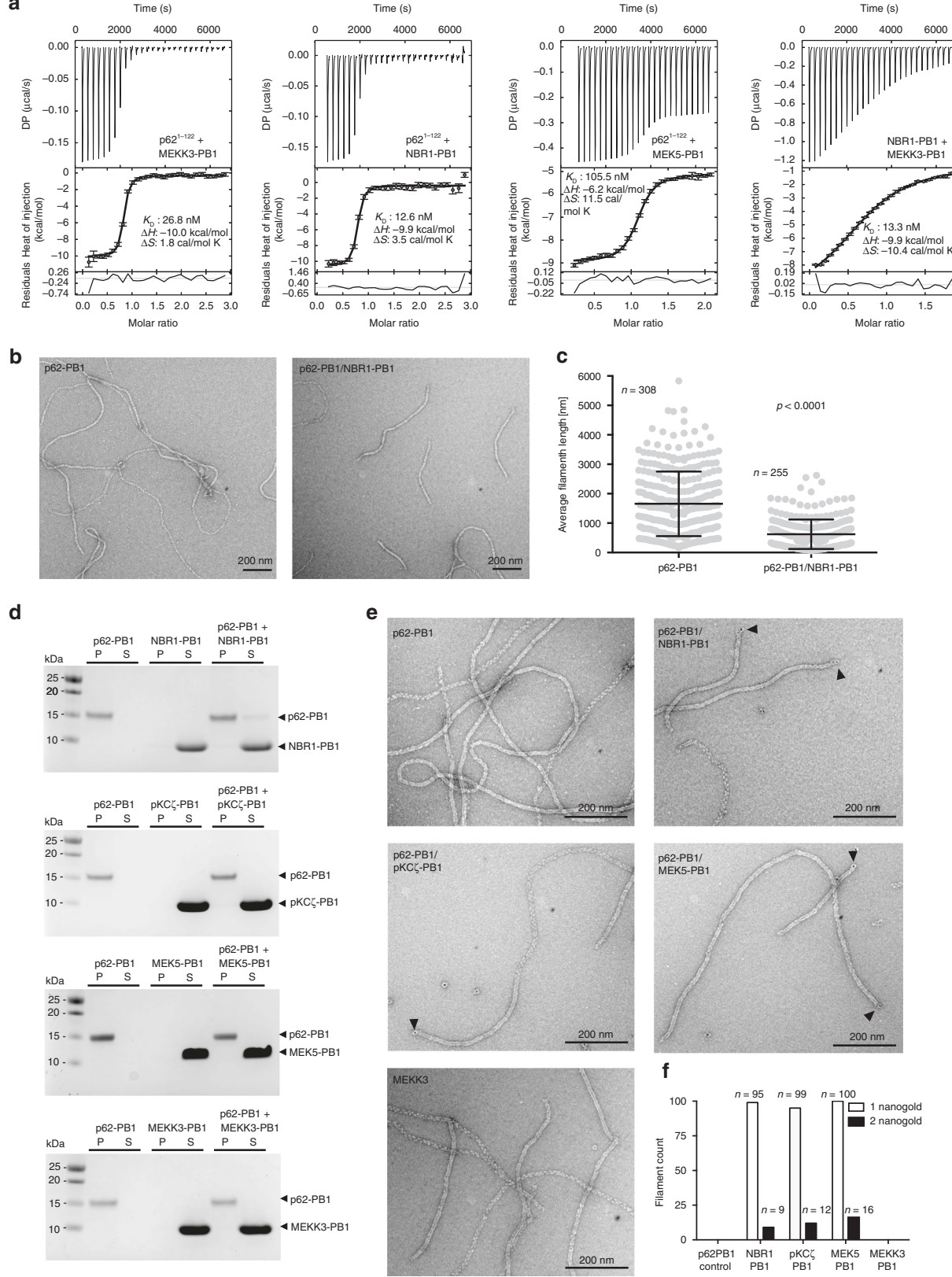

## Discussion

The PB1 domain is a common interaction module present in all kingdoms of life and found in various proteins involved in membrane trafficking, redox regulation, cell division, as well as in signaling. In this study, we focused on the structure in addition to the biological and functional relevance of the p62-PB1 domain in the context of polymeric assemblies. The overall ubiquitin-like fold of the PB1 domain has been determined, and different interface types through acidic and basic patches have been identified in earlier studies[2,5]. Our cryo-EM structures of filamentous p62 and AtNBR1–PB1 assemblies revealed that the presence of a tandem arginine sequence in the basic motif of type

**Fig. 4 Interactions of p62-PB1 with other PB1 domain proteins. a** Quantitative determination of PB1-binding affinities by isothermal titration calorimetry. Data represent mean and standard deviations from three independent experiments. **b** Representative electron micrographs of negatively stained p62-PB1$^{1–122}$ (left) incubated with human NBR1–PB1 (right). **c** Quantification of lengths of P62-PB1$^{1–122}$ filaments before and after incubation with NBR1–PB1. Source data are provided as a Source Data file. **d** Co-sedimentation assays of p62-PB1$^{1–122}$ with NBR1-PB1, PKCζ-PB1, MEK5-PB1, and MEKK3-PB1 (S = supernatant; P = pellet). Control experiments of p62-PB1$^{1–122}$ and the respective PB1 interactor alone are also shown. Source data are provided as a Source Data file. **e** Representative electron micrographs of negatively stained p62-PB1$^{1–122}$ with nanogold-labeled NBR1–PB1, PKCζ–PB1, MEK5–PB1, or MEKK3–PB1. **f** Quantification of p62-PB1$^{1–122}$ filaments displaying one or two nanogold-labeled PB1 interaction domains. Source data are provided as a Source Data file.

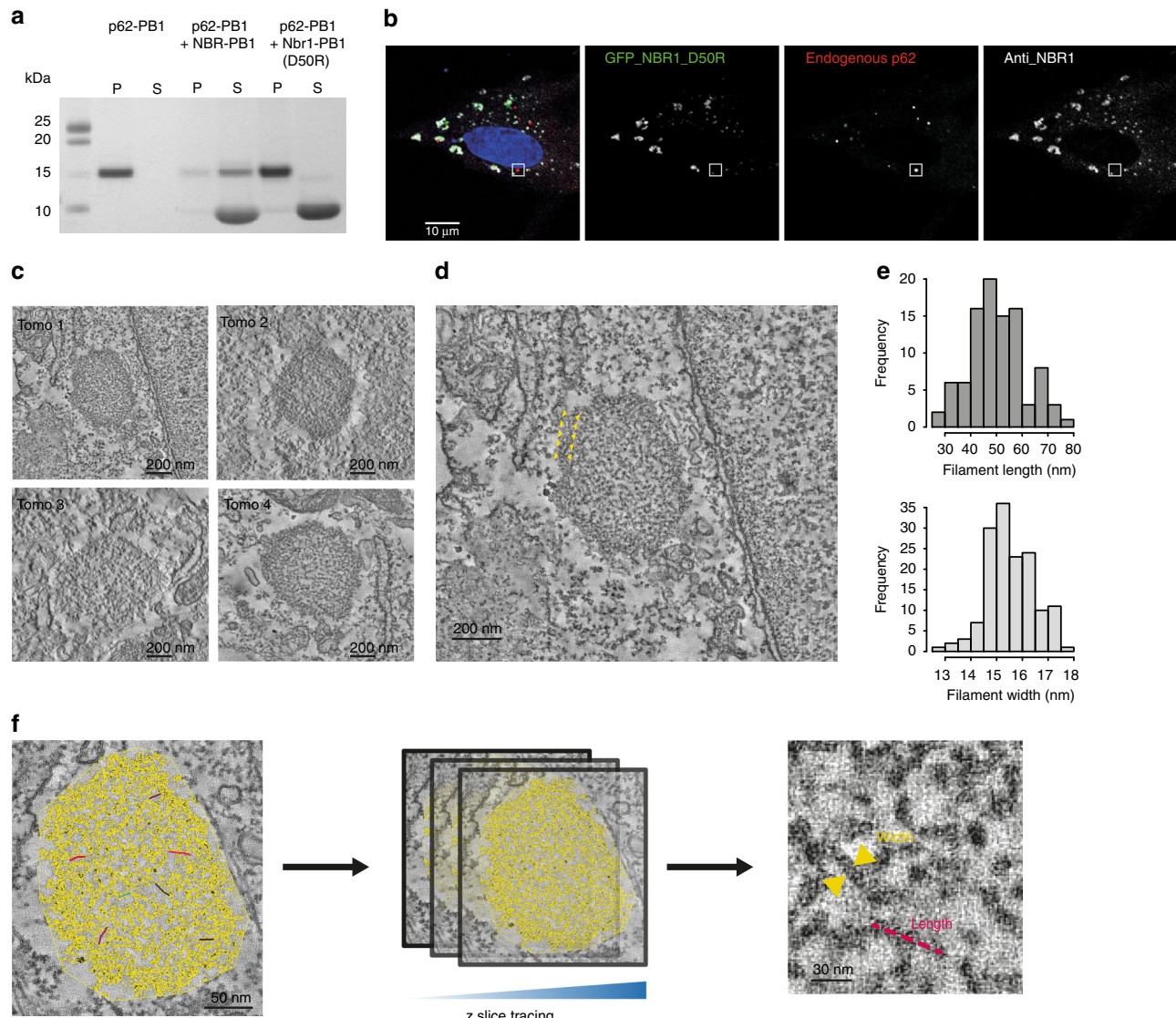

**Fig. 5 CLEM visualization of p62 bodies in cells. a** Effect of human NBR1-D50R mutation on p62 filaments. SDS-PAGE analysis of pelletation assay showing that p62-PB1 filaments are not disrupted by NBR1–PB1 with a D50R mutation (P = pellet; S = supernatant). Source data are provided as a Source Data file. **b** Representative confocal fluorescence images showing NBR1 (green) and endogenous p62 (red) in RPE1 cells. Co-localization analysis of fixed RPE1 cells stably expressing NBR1(D50R) shows no overlap of NBR1(D50R) with p62 bodies. **c** Representative electron tomogram slices of p62 bodies localized by CLEM. **d** Enlarged view of a representative tomogram slice from the highlighted p62 body in (**b**) reveals the filament-like meshwork of p62 bodies. Note the apparent phase separation of the p62 body from the cytosol. The ring of increased density surrounding the bodies is indicated by yellow arrows. **e** Distribution of estimated filament length and width from tracing in thresholded tomograms. Source data are provided as a Source Data file. **f** Schematic illustration of width and length measurements performed in thresholded tomograms (yellow pixels).

AB interfaces is required to stabilize a polymeric assembly. Although the exact composition of the interface between opposed and electrostatically complementary surfaces is distinctly different for the two PB1 assemblies, the main functional acidic and basic residues including the essential double arginine finger are conserved (Fig. 3). Furthermore, we observed that the propagation of

the helical rung is also distinctly different in p62 and AtNBR1 assemblies, with small changes in primary structure giving rise to large differences in quaternary structure. This property has been characterized in other sequence-related helical systems[28]. Interestingly, we also found that the polymeric PB1 domain structures of human p62 and AtNBR1 are assembled from a common helical

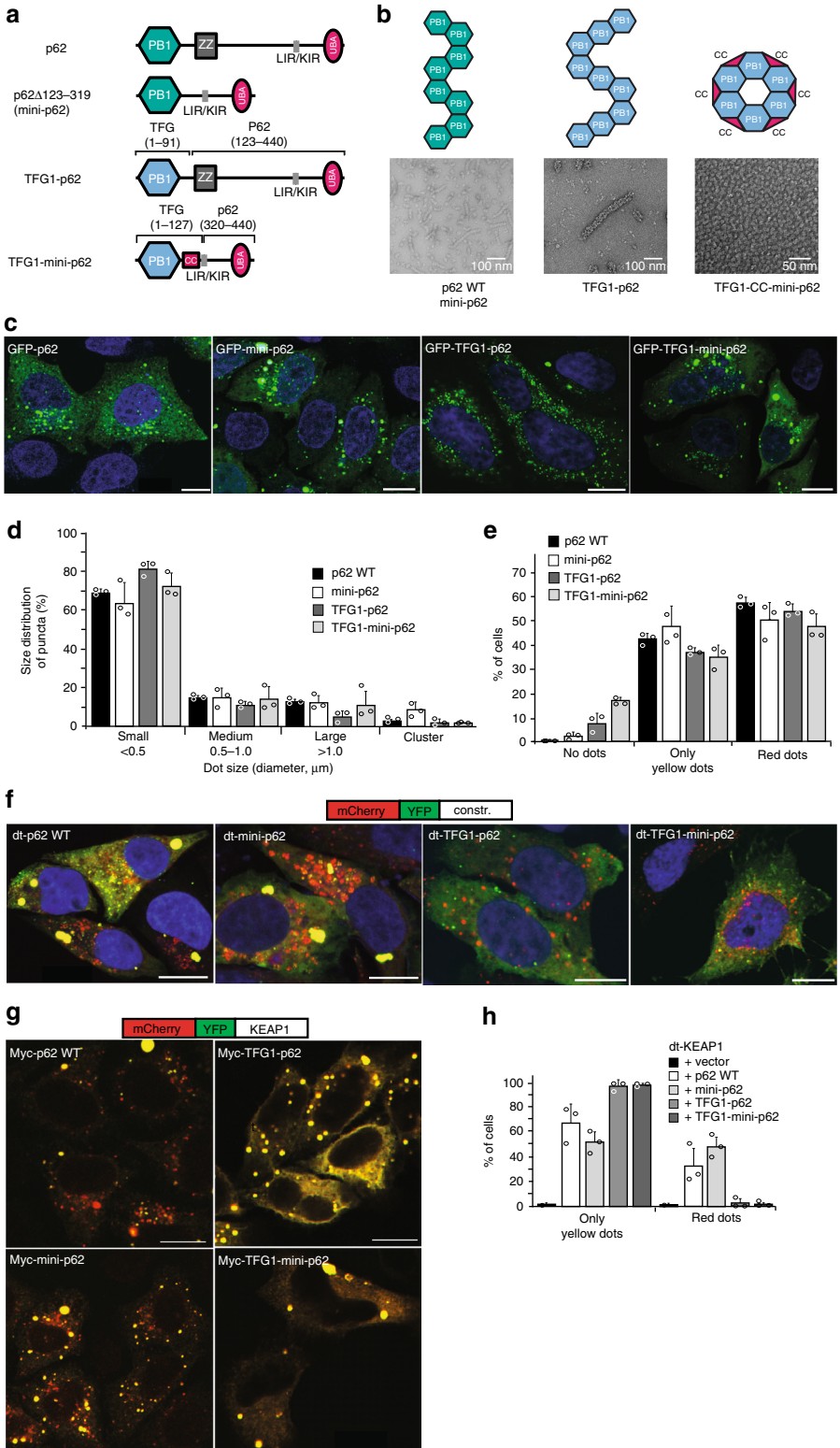

rung into two morphologically distinct organization types, i.e., in the form of differently organized helical rungs. We speculate that this observed plasticity of assembling a common helical rung is a consequence of flexibility in forming the longitudinal PB1–PB1 interactions in the loop regions. As the constructs used here for structure determination and cellular assays were limited to PB1 domains of AtNBR1 and p62, the relevance and functional consequences of these different morphological arrangements within

cellular polymeric assemblies remain open. Full-length p62 was shown to be flexible, and at this stage too disordered to be amenable to 3D reconstructions[8]. In line with our previous analysis, the PB1 domain directs the C-terminus either to the outside or the inside of the helical assembly, depending on the exact helical arrangement. It is possible to envision that different morphological arrangements affect the availability of critical interaction motifs outside the PB1 domain, i.e., LIR and KIR motifs as well as the UBA domain.

**Fig. 6 Cellular assays of p62 polymeric state. a** Schematic illustration of used p62 constructs and chimeras with p62-PB1 (green) and TFG1–PB1 (blue). **b** Representative, negatively stained electron micrographs of purified p62 constructs and chimeras from (**a**), including illustration of polymeric and oligomeric forms observed by negative staining electron microscopy. **c** Confocal fluorescent images of HeLa p62 (KO) cells expressing GFP-tagged constructs and chimeras. All examined constructs form punctate structures. **d** Quantification of the number of p62 bodies forming dots of various sizes. **e** Quantification of cells displaying yellow and red dots in (**f**). **f** Representative confocal fluorescence images of HeLa p62 (KO) cells expressing mCherry-YFP-tagged (dt-tagged) p62 constructs and chimeras. The appearance of red puncta (as an indicator of lysosomal localization) for all constructs indicates that all constructs and chimeras can be processed by autophagy. Punctae were counted and classified based on more than 100 cells in each condition in three independent experiments. **g** Representative confocal fluorescence images of HeLa p62 (KO) cells expressing the respective p62 constructs and chimeras, as well as mCherry-YFP-tagged KEAP1. **h** Statistics of appearance of lysosome-localized and cytosolic dots for mCherry-YFP-tagged KEAP1. The error bars in **d**, **e**, and **h** represent standard deviations of the mean.

Previous studies showed that purified full-length p62 can also form helical filaments[7,8]. The existence of these assembly structures inside of cells, however, had not been demonstrated. Therefore, we used the CLEM technique to identify and visualize the ultrastructural organization of p62 found in large clusters known as p62 bodies. Image analysis confirmed that p62 bodies consist of a meshwork of short filamentous structures. The principal dimension of the observed structures is consistent in width and length with previous measurements in vitro[8]. The structures are compatible with recently observed aggregates of p62 in brain neurons and neuroepithelial cells[20]. Due to the limited length and flexibility, p62 filaments pack loosely into a spheroid-shaped, meshwork-like superstructure. The observed bodies with average dimensions below micrometers in size aggregate in structures that appear morphologically separated from the cytosol (Fig. 5), suggestive of phase separation as observed previously in reconstitution experiments[21,22]. The observed body structures of hundreds of nanometers are also significantly larger than individual filaments with on average 30 nm length. When organized in such large superstructures, p62 bodies are more similar in dimension to typical molecular cargo, such as protein aggregates, viruses, and organelles when compared with receptor oligomers or filament assemblies alone.

The organization of p62 in filamentous assemblies has direct functional consequences for the interaction with a series of binding partners in the context of autophagy as well as signaling. It has been demonstrated that a polymeric organization of p62 can enhance low-affinity interactions to highly avid interactions[13]. In addition, using p62-interacting PB1 domains from MEK5, PKCζ, and MEKK3 kinases, we show that p62 polymeric assemblies can be capped on one end or dissociate into smaller, ring-like structures. The intact p62 filaments occlude the bulk of PB1 interaction sites that are accessible in its monomeric state[5] (Fig. 4). Conversely, we show that end binding of NBR1 to p62 filaments leads to disassembly and shortening, which can thereby modulate the length of the filamentous structure. As NBR1 binding has been shown to promote p62 body formation in vitro[21,22] to co-localize with p62 bodies in vivo[29], we hypothesize that this filament-end interaction by NBR1 cross-links shorter filaments more effectively into larger structures and thereby also affects the size of p62 bodies in cells. We speculate that other interactors have similar effects on the size and dynamics of p62 bodies as they may occur in phase separation processes. The size of bodies will also control the availability of interaction sites. The here presented structures and interaction studies of PB1–p62 filaments reveal a series of regulation mechanisms that are critical in the functional context of p62's action in autophagy and signaling.

In order to understand how the assembly state, the specific symmetry, and subunit arrangement of this state affect p62's biological function, we tested a series of chimera variants of p62 for their efficiency with regard to cargo uptake and autophagic degradation in the cell. The experiments showed that polymeric

as well as oligomeric ring-like scaffolds from related PB1 domains fused to the C-terminal functional domains of p62 can be taken up by the autophagy machinery almost as efficiently as WT-p62. Interestingly, this is not the case for variants of p62 that are monomeric and diffuse in the cytosol[12]. Our results suggest that structures organized in larger oligomeric clusters are sufficient to mediate self-disposal of p62 (Fig. 6), presumably due to increased avidity of accessible LIR motifs and UBA domains. The specific uptake of the model cargo KEAP1, however, could only be accomplished by WT-p62 and mini-p62 retaining the structural context of native p62 assemblies. Other TFG1-PB1-p62 chimera polymers were not capable of transferring KEAP1 to the lysosome. The dependency of the native p62-PB1 domain for filament assemblies and KEAP1 degradation was further illustrated by the monomeric double arginine finger (R21A/R22A) mutant of p62. This mutant was completely diffusely localized, not degraded by autophagy, and unable to mediate degradation of KEAP1 by autophagy. In conclusion, larger p62 assemblies, including ring-like structures and filaments, are essential for disposal of autophagy cargo. Moreover, the precise structural context of the filament assembly is affecting the ability to degrade KEAP1-containing aggregates, and possibly other p62-specific cargoes.

## Methods

**Protein purification.** AtNBR1 residues 1–94 (NBR1–PB1), p62 residues 1–122 (p62-PB1), and TFG1 residues 1–95 (TFG1–PB1) were cloned into a pETM44 expression vector containing a N-terminal His₆ tag, followed by a maltose-binding protein (MBP) tag and a recognition sequence for 3C protease. Proteins were expressed in *E. coli* BL21 (DE3) (obtained from Protein Expression and Purification Core Facility EMBL) using auto-induction in lactose-containing media[30]. After 18 h, cells were harvested by centrifugation, resuspended in lysis buffer (50 mM HEPES, pH 8.0, 0.5 M NaCl, 0.05 mM TCEP, and 0.1% (v/v) Triton X-100), and lysed by three cycles of rapid freeze–thawing in liquid nitrogen. After removal of cell debris by centrifugation, recombinant proteins were purified by Ni-NTA affinity chromatography, and diafiltrated into 50 mM HEPES, pH 7.5, 0.1 M NaCl, and 0.05 mM TCEP followed by proteolytic cleavage of the His₆/MBP by incubation with 1:200 mol/mol 3 C protease at ambient temperature. After 1 h, the cleavage solution was incubated with Talon resin (Clontech) for 15 min, and the resin subsequently sedimented by centrifugation. The supernatant contained the respective PB1 domains in high purity. p62 residues 1–122 (p62-PB1) were cloned into pOPTM and expressed as an MBP fusion protein in *E. coli* BL21 (DE3) using auto-induction (Studier 2005). NBR1 residues 1–85 (NBR1–PB1), pKCζ residues 11–101 (pKCζ–PB1), MEK5 residues 5–108 (MEK5–PB1), and MEKK3 residues 43–127 (MEKK3–PB1) were cloned into the pETM11 containing an N-terminal His₆ tag followed by a recognition sequence for TEV protease. Proteins were expressed in *E. coli* BL21 (DE3) using auto-induction (Studier 2005). For the gold-labeling experiments, the His₆ tag was not removed to allow binding of 5 nm Ni-NTA-Nanogold® (Nanoprobes). For consistency, the His₆ tag was also kept on the proteins for the co-pelletting assay.

**Thermal unfolding assays.** Thermal denaturation assays were performed essentially as described previously[31]. Briefly, protein was dialyzed into 15 mM HEPES (pH 7.5), 150 mM NaCl for pH screening, or 100 mM HEPES (pH 7.5) for ionic strength screening. All additives were dissolved in 50 mM HEPES (pH 7.5). A volume of 12.5 μl of a solution containing 500 ng of protein was diluted in H₂O with 5x Sypro Orange (Sigma-Aldrich) and immediately mixed with an equal volume of assay condition. All conditions were assessed in triplicate. Fluorescence increase was monitored on a MyiQ real-time PCR instrument (BioRad). Assays were performed over a temperature range of 15–90 °C using a ramp rate of 1 °C

min$^{-1}$ in steps of 0.5 °C. Fluorescence data from triplicate measurements were baseline corrected individually, and unfolding curves were normalized to maximum fluorescence to give fractional denaturation curves. The apparent $T_m$ was determined as the inflection point of a sigmoidal fit to the normalized fluorescence signal using a customized routine in R.

**Quantification of PB1-binding affinities.** Isothermal titration calorimetry (ITC) experiments were carried out with a VP-ITC system (MicroCal). Experiments were performed at 25 °C in 10 mM HEPES (pH 7.5), 150 mM NaCl. Purified p62$^{1-122}$ D69A/D73A was placed in the reaction cell at a concentration of 5–20 μM with either MEK5 or NBR1 at a concentration of 25–100 μM in the injection syringe. Injections of 10 μl of syringe solution were performed at 4-min intervals. Integration of the raw thermogram data, baseline correction, and data processing were performed with the NITPIC[32] and SEDPHAT[33] software packages. The data were corrected by the heat of injection calculated from the basal heat remaining after saturation. A one-site binding mode was used to fit the data using a nonlinear least-squares algorithm[34]. The values reported are the mean of three independent measurements, and errors represent the corresponding standard deviation.

**Co-pelleting assay.** Co-pelleting assay was performed according to the F-actin binding co-sedimentation assay from Cytoskeleton Inc. In brief, p62-PB1$^{1-122}$, potential binding partner, or p62-PB1$^{1-122}$ together with potential binding partner, were incubated for 1 h on ice followed by centrifugation at 49,000 $g$, 4 °C for 30 min in a TLA-100 rotor. The pellet and supernatant were assayed by SDS-PAGE and stained with Coomassie.

**Negative staining EM and filament-length measurements.** p62-PB1$^{1-122}$ was incubated with different binding partners for 1 h on ice followed by 30 min of incubation with 5 nm Ni-NTA-Nanogold® (diluted 1:25). Excess nanogold was removed through pelletation of filaments by ultracentrifugation at 49,000 $g$, 4 °C for 30 min in a TLA-100 rotor, and the pellet fraction was resuspended in 20 mM HEPES, pH 8, 50 mM NaCl. The sample (3.6 μl) was applied to a glow-discharged carbon-coated EM grid and blotted according to the side-blotting method[35]. Grids were imaged using a Morgagni 268 transmission electron microscope (FEI) operated at 100 kV with a side-mounted 1 K CCD camera. Filament lengths for p62-PB1$^{1-122}$ and p62-PB1$^{1-122}$/HsNBR1$^{1-85}$ were measured using Fiji[36], and statistical analysis was done using a two-tailed unpaired $t$ test with Welch's correction in GraphPad Prism 6.0.

**Pull-down assay.** MBP-tagged AtNBR1–PB1$^{1-94}$ and mutant AtNBR1 residues 1–94 (AtNBR1–PB1) were expressed as described above and buffer-exchanged into 15 mM Tris (pH 7.5), 150 mM NaCl. For the AtNBR1 pull-down experiments, 50 μl of amylose resin (NEB) was incubated for 10 min with MBP–AtNBR1–PB1, followed by 5-min incubations with a 4:1 molar excess of mutant AtNBR1–PB1. Beads were washed with 15 mM HEPES (pH 7.5), 500 mM NaCl, eluted with 15 mM HEPES (pH 7.5), 150 mM NaCl, and 30 mM maltose, and fractions were analyzed by SDS-PAGE. p62, TFG1-mini-p62, and TFG1-p62 were purified as described and applied to size-exclusion chromatography on a Superdex 75 16/60 in 20 mM Tris (pH 8), 100 mM NaCl. Human KEAP1–DC$^{309-624}$ was cloned into pET-28a(+) with an N-terminal 6×His tag, purified on Ni-NTA resin, and buffer-exchanged into 20 mM Tris (pH 8), 100 mM NaCl. Approximately 100 μl of amylose resin (NEB) was incubated for 30 min with either of the MBP-containing proteins at room temperature, followed by 30-min incubations with a 3:1 molar excess of KEAP1–DC$^{309-624}$. Beads were washed with 20 mM Tris (pH 8), 1 M NaCl, and eluted with 20 mM Tris (pH 8), 100 mM NaCl, and 20 mM maltose. Fractions were analyzed by SDS-PAGE.

**Electron cryo-microscopy.** For AtNBR1$^{1-94}$, a total of 3.0 μl of 0.4 mg ml$^{-1}$ AtNBR1–PB1 was applied to glow-discharged C-flat grids (CF-1.2/1.3–2 C, 400-mesh holey carbon on copper; Protochips) on a Leica GP2 vitrification robot (Leica, Germany) at 95% humidity and 25 °C. The sample was incubated for 10 s on the grid before blotting for 2 s from the back side of the grid and immediately flash-frozen in liquid ethane. Micrographs were acquired at 300 kV using an FEI Titan Krios (Thermo Fisher Scientific) equipped with a Falcon II direct detector at a magnification of 59,000, corresponding to a pixel size of 1.386 Å at the specimen level. Image acquisition was performed with EPU Software (Thermo Fisher Scientific), and micrographs were collected at an underfocus varying between 0.5 and 4.5 μm. We collected a total of seven frames accumulating to a dose of 14 e$^-$ Å$^{-2}$ over 0.82 s. In total, 742 micrographs were acquired, of which we selected 684 for further processing after discarding micrographs that did not show Thon rings exceeding 6 Å.

For p62$^{1-122}$, L-type filaments were enriched by the following procedure: 0.2 mg of p62-PB1 (100 μl) was ammonium sulfate precipitated (25% v/v) and incubated o/n at 4 °C. The sample was spun at 17,000 $g$ for 15 min at 4 °C, and the pellet was resuspended in 50 mM TRIS (pH 7.5), 100 mM NaCl, and 4 mM DTT. This ammonium sulfate precipitation was repeated a second time. In the final step, the sample was centrifuged at 49,000 $g$ for 45 min at 4 °C, and the pellet resuspended in 25 μl. A total of 3.6 μl of the resulting p62-PB1$^{1-122}$ solution was applied to glow-discharged Quantifoil R2/1 Cu 400-mesh grids on a Vitrobot Mark IV (Thermo

Fisher Scientific) at 10 °C and 100% humidity. The sample was blotted for 5 s from both sides, and flash-frozen in liquid ethane after a drain time of 1 s. Micrographs were acquired at 300 kV using a FEI Titan Krios (Thermo Fisher Scientific) with a K2 Summit detector (Gatan, Inc.), a pixel size of 1.04 Å, and an underfocus ranging from 0.5 to 2.5 μm. In total, 40 frames were collected in counting mode with a dose rate of 4.5 e$^-$ Å$^{-2}$ s and a total dose of 40 e$^-$ Å$^{-2}$. In total, 2277 micrographs were automatically collected, and 856 micrographs without ice contamination or carbon chosen for further processing.

**Image processing.** For the AtNBR1 dataset, movie frames were aligned using MOTIONCORR[37]. The resulting frame stacks and integrated images (total frame sums) were used for further processing. The contrast-transfer function of the micrographs was determined with CTFFIND4[38] using the integrated images. Helix coordinates were picked using e2helixboxer.py from the EMAN2 package[39]. Initially a subset of 100 images was selected for preliminary processing in SPRING[40]. Briefly, overlapping helix segments of 350 × 350 Å dimensions were excised from the frame-aligned images with a mean step size of 60 Å using the SEGMENT module in SPRING. In-plane rotated, phase-flipped segments were subjected to 2D classification by k-means clustering as implemented in SPARX[41]. During a total of five iterations, the segments were classified and iteratively aligned against a subset of class averages chosen based on the quality of their power spectra. Class averages revealed two distinct helix types referred to as S-type and L-type. We determined the helical symmetry for the L-type helices by indexing of the power spectra obtained from the 2D classification. The final symmetry parameters were determined with a symmetry search grid using SEGMENTREFINE3DGRID. For 3D refinement and reconstruction, the excised segments were convolved with the CTF and no in-plane rotation was applied prior to reconstruction. Starting from the symmetry parameters obtained for the L-type helix, symmetry parameters of the S-type helix were refined. The maximum of the mean cross-correlation peak between computed and experimental power spectra was found at a pitch of 70 Å, 11.55 units per turn for the two-start L-type helix, and a pitch of 68.2 Å, 11.55 units per turn for the one-start S-type helix. Using the refined symmetry parameters, we performed a competitive high-resolution multi-model structure refinement using all 684 images with a final resolution of 4.5/3.9 Å and 5.0/4.4 Å (FSC 0.5/0.143)[42] for the two-start (L-type) and one-start (S-type) helix reconstructions (Table 2).

For the p62$^{1-122}$ data set, movie frames were aligned in RELION3[43] using 5 × 5 patches. The contrast-transfer function of the micrographs was determined with Gctf[44]. Helix coordinates were automatically picked in RELION3 and segments extracted with a step of 22.5 Å, binning 2, and an unbinned box size of 256 pixels. 2D classification with 100 classes was performed and classes were selected that showed secondary structure features. Two separate subsequent 2D classifications were performed with two distinct groups of 2D classes belonging to an S-type and L-type pattern. Using SEGCLASSRECONSTRUCT from the SPRING package[40] a series of putative helical symmetry solutions could be obtained. In addition to running a series of refinements with these symmetry solutions, a C1 reconstruction provided additional hints for symmetry parameters. Imposition of wrong symmetry parameters led to smeared density features, whereas only the correct symmetries for both filament types led to recognizable high-resolution side-chain features. Helical symmetry was automatically refined in RELION to 77.3° helical rotation and 4.8 Å rise for the S-type and 26.5° rotation and 9.8 Å rise for the L-type, respectively (Table 2). Focussed refinement was performed using a mask covering the central 25% of the filament along the helical axis. This approach improved the resulting resolution for the L-type, but not for the S-type. The final resolution was estimated at 3.5 Å and 4.0 Å using the FSC and the 0.143 criterion cutoff[42], for the L- and S-types, respectively.

**Atomic model building and refinement.** For visual display and model building, the AtNBR1 EM density map of the individual reconstructions was initially filtered to 3.9 Å and 4.4 Å, respectively, and sharpened with a B-factor of −200 Å$^2$. The AtNBR1–PB1 subunit model was built into the 3.9 Å density map of the L-type arrangement de novo in COOT[45]. Residues 81–85 could not be built de novo due to weak density, but were added based on the high-resolution crystal structure obtained in this study, which showed good agreement with the weak density. For the p62-PB1 (3–102) map, the NMR structure from *rattus norvegicus* (PDB ID 2kkc [https://doi.org/10.2210/pdb2kkc/pdb]) was rigid-body fitted into the RELION-postprocessed density of the L-type filament and then manually adjusted to the human sequence in COOT[45]. The models were expanded using helical symmetry, and a nine-subunit segment was excised to serve as a refinement target, taking into account interactions along the azimuthal propagation and lateral interactions along the helical axis. Following real-space refinement in PHENIX[46], we used model-based density scaling[47] to generate locally sharpened maps and completed the model in COOT followed by further iterations of real-space refinement. The final monomer atomic model from the L-type arrangement was rigid-body fitted into the S-type density, and refinement of the model was performed as described above (Table 3).

**X-ray crystallography.** Crystals of AtNBR1$^{1-94}$ carrying a D60A/D62A mutation were grown using hanging drop vapor diffusion at 292 K by mixing equal volumes of 11 mg ml$^{-1}$ protein and reservoir solution. Within 10 h, crystals appeared as

needle clusters in 0.085 M MES (pH 6.5), 18.2% (w/v) PEG20000. Isolated needles ($10 \times 2 \times 4$ µm) were obtained by streak seeding with a cat whisker into 0.1 M MES (pH 6.5), 18–20% (w/v) PEG20000, or 0.1 M sodium cacodylate (pH 6.5), 0.2 M $(NH_4)_2SO_4$, and 30–33% PEG8000. For cryo-protection, crystals were soaked in the crystallization condition supplemented with 15% (v/v) glycerol. Diffraction data were collected on the ID23-2 microfocus beamline at the European Synchrotron Radiation Facility (ESRF) and processed with XDS[48] and AIMLESS[49]. Initial attempts to solve the crystal structure using the cryo-EM atomic model were unsuccessful. The crystal structure was solved using molecular replacement using the monomer density from the L-type cryo-EM reconstruction as the search model. Briefly, the monomer density was obtained by cutting out density extending 4.5 Å beyond the atomic coordinates. The extracted map segment was centered in a P1 unit cell extending over three times the maximum map dimension, converted to structure factors using a in-house, customized CCTBX[50] routine, and used for automated molecular replacement in PHASER[51]. The top-scoring solution had a translation function Z score of 16.5. Henderson–Lattmann coefficients were generated from the figure of merit (FOM) obtained from the PHASER solution and employed for phase extension using the high-resolution X-ray crystallographic data by density modification in RESOLVE[52], yielding excellent electron density. Using the 1.9 Å data, the model was built using Arp/Warp[53] and completed manually in COOT. Table 1 summarizes data collection and refinement statistics.

**Correlative light and electron microscopy.** For CLEM, RPE1 cells (ATCC CRL-4000) were transiently transfected with pDest-EGFP-NBR1(D50R)[29] and grown on photo-etched coverslips (Electron Microscopy Sciences, Hatfield, USA). Cells were fixed in 4% formaldehyde, 0.1% glutaraldehyde/0.1 M PHEM (80 mM PIPES, 25 mM HEPES, 2 mM $MgCl_2$, and 10 mM EGTA, pH 6.9), for 1 h. The coverslips were then washed in PBS containing 0.005% saponin and stained with the indicated primary antibodies for 1 h (rabbit anti-p62 (MBL, PM045), mouse anti-NBR1 (Santa Cruz, #sc-130380)), washed three times in PBS/saponin, stained with secondary antibodies (from Jackson ImmunoResearch Laboratories) for 1 h, washed three times in PBS, and shortly rinsed in water. The cells were mounted with Mowiol containing 2 µg ml$^{-1}$ Hoechst 33342 (Sigma-Aldrich). Mounted coverslips were examined on a Zeiss LSM780 confocal microscope (Carl Zeiss MicroImaging GmbH, Jena, Germany) utilizing a Laser diode 405–30 CW (405 nm), an Ar-Laser Multiline (458/488/514 nm), a DPSS-561 10 (561 nm), and a HeNe laser (633 nm). The objective used for confocal microscopy was a Zeiss plan-Apochromat 63×/1.4 Oil DIC III. Cells of interest were identified by fluorescence microscopy and a Z stack was acquired. The relative positioning of the cells on the photo-etched coverslips was determined by taking a DIC image. The coverslips were removed from the object glass, washed with 0.1 M PHEM buffer, and fixed in 2% glutaraldehyde/0.1 M PHEM for 1 h. Cells were postfixed in osmium tetroxide, stained with tannic acid, dehydrated stepwise to 100% ethanol, and flat-embedded in Epon. Serial sections (~100–200 nm) were cut on an Ultracut UCT ultramicrotome (Leica, Germany), collected on formvar-coated mesh grids, and post-stained with lead citrate.

**Electron tomography from cellular sections.** Samples were observed using a FEI Talos F200C electron microscope (Thermo Fisher Scientific). Image series were taken between −60° and 60° with 2° increment. Single-tilt or double-tilt series (as indicated in the text above) were recorded with a Ceta 16 M camera. Single-axis tomograms were computed using weighted back projection, and when applicable, merged into a dual-axis tomogram using the IMOD[54] package. Display and animation of segmentation of tomograms were performed using a scripted workflow in ImageJ[36] and IMARIS.

**Autophagy and p62 turnover assays.** The following antibodies were used: mouse anti-Myc antibody (Cell Signaling, Cat. #2276#, 1:8000 for western blots and 1:5000 for confocal imaging); rabbit anti-GFP antibody (Abcam, ab290, 1:5000); guinea pig anti-p62 antibody (Progen, Cat. #Gp62-C#, 1:5000); rabbit anti-actin antibody (Sigma, Cat. #A2066#, 1:1000); Alexa Fluor® 647-conjugated goat anti-mouse IgG (A21236, 1:1000); HRP-conjugated goat anti-mouse IgG (1:5000); goat anti-rabbit IgG (1:5000); goat anti-guinea pig IgG (1:5000).

**Generation of HeLa cells KO for p62 and stable cell lines.** To generate CRISPR/Cas9 p62 gRNA plasmid, sense and antisense p62 gRNA was annealed and then inserted into plasmid pX330 (Ref PMID: 23287718). For generation of CRISPR/Cas9 p62 KO cells, ~30,000 HeLa cells (ATCC CCL2) were seeded per well into 24-well plates and transfected with plasmid pX330 p62 gRNA using Metafectene Pro (Biontex T040). For clonal selection, cells were treated with 500 ng ml$^{-1}$ of puromycin 24 h after transfection for 48–72 h. Later, single cells were sorted into a 96-well plate using FACS (fluorescence-activated cell sorting). These clones were allowed to grow for 7–10 days before screening for KO using immunoblotting. The following sense 5′-CACCGTCATCCTTCACGTAGGACA-3′ and antisense 5′-AAACTGTCCTACGTGAAGGATGAC-3′ gRNAs were used.

HeLa FlpIn T-Rex p62 KO cells[55] were used to make stable cell lines expressing GFP-p62 or GFP-p62 R21A/R22A. First, p62 and p62 R21A/R22A were transferred into the destination vector pDest-FRT/TO-GFP-C1[56] by Gateway LR recombination reactions. Then stable cell lines were made using the manufacturer's instructions (Invitrogen, V6520-20). Briefly, the HeLa FlpIn T-Rex p62 KO cell line was transfected with pDest-FRT/TO-EGFP-p62 or pDest-FRT/TO-EGFP-p62 R21A/R22A. Forty-eight hours post transfection, colonies of cells with the gene of interest integrated into FRT site were selected using 200 µg/ml of Hygromycin (Calbiochem, 400051). To induce expression of the gene of interest, 1 µg/ml of tetracycline was added for 24 h. Analyses of degradation of EGFP-p62 or EGFP-p62 R21A/R22A by flow cytometry were done as previously described[57].

**Construction of plasmids.** The gateway entry clones pENTR-p62, pENTR-p62 R21A/R22A, and pENTR-p62 Δ123–319 (mini-p62) have been described previously[2]. pENTR-p62 Δ123–319 was made by deletion of pENTR-p62. TFG1-p62 fusion constructs were produced by InFusion PCR. To subclone the TFG1-p62 fusion constructs into an ENTRY vector, a NcoI site was inserted into the start codon of p62 in pENTR-p62, creating pENTR-p62$_{CCATGG}$. The start codons in TFG1-p62 (AJD152) and TFG1-mini-p62 (AJD157) already have NcoI sites, and there is an additional NcoI site close to the end of the p62 cDNA sequence in pENTR-p62, TFG1-p62, and TFG1-mini-p62. To replace wild-type p62 of pENTR-p62$_{CCATGG}$, TFG1-p62 and TFG1-mini-p62 (AJD152 and AJD157) were subcloned as NcoI fragments into pENTR-p62CCATGG cut with NcoI, creating pENTR-TFG1-p62 and pENTR-TFG1-mini-p62, respectively. Gateway LR recombination reactions were performed as described in the Gateway cloning technology instruction manual (Thermo Fisher Scientific, 11791020). Gateway expression clones pDest-Myc-p62, pDest-EGFP-p62, and pDest-mCherry-EGFP-KEAP1 have been described previously[2,16]. pDest-TFG1-Myc-p62 Δ123–319, pDest-Myc-TFG1-p62, and pDest-Myc-TFG1-mini-p62 were made by Gateway LR reactions using destination vector pDest-Myc (mammalian expression of N-terminal Myc-tagged proteins[2]). pDest-EGFP-p62 Δ123–319, pDest-EGFP-TFG1-p62, and pDest-EGFP-TFG1-mini-p62 were made using destination vector pDest-EGFP-C1 (mammalian expression of N-terminal EGFP-tagged proteins[2]). pDest-mCherry-EYFP-p62, pDest-mCherry-EYFP-p62 Δ123–319, pDest-mCherry-EYFP-TFG1-p62, and pDest-mCherry-EYFP-TFG1-mini-p62 were made using destination vector pDest-mCherry-EYFP[58] (mammalian expression of N-terminal mCherry-EYFP double-tagged proteins).

**Cell culture and transfections.** HeLa p62 KO cells were cultured in Eagle's minimum essential medium with 10% fetal bovine serum (Biochrom AG, S0615), non-essential amino acids, 2 mM L-glutamine, and 1% streptomycin–penicillin (Sigma, P4333). For transfection the same media was used but without 1% streptomycin–penicillin. Cells were fixed in 4% PFA for 20 min at room temperature. For immunostaining, cells were permeabilized with cold methanol for 5 min at room temperature, blocked in 3% goat serum/PBS, and incubated at room temperature with antibodies. For DNA staining 1:4000 dilution was used in PBS of DAPI (Thermo Fisher Scientific; pr.66248). Samples were mounted using Mowiol 4-88 (Calbiochem 475904). Cells were examined using a Zeiss LSM780 or LSM800 microscope with a 63 × 1.4 oil objective or a Leica TCS SP8 confocal microscope, 40 × 1.3 oil objective.

**Western blot analyses.** Transfected HeLa p62 KO cells were harvested in 50 mM Tris, pH 7.4, 2% SDS, and 1% glycerol. Cell lysates were cleared by centrifugation, and supernatants resolved by SDS-PAGE and transferred to Hybond-ECL nitrocellulose membrane (GE Healthcare). The membrane was blocked with 5% nonfat dry milk in PBS-T, incubated with primary antibody overnight, and HRP-conjugated secondary antibody for 1 h at room temperature. Proteins were detected by immunoblotting with a chemiluminescence Luminol kit (SC-2048, Santa Cruz Biotechnology) using a LumiAnalyst Imager (Roche Applied Sciences).

**Reporting summary.** Further information on research design is available in the Nature Research Reporting Summary linked to this article.

## Data availability

The PDB accession number for the atomic coordinates and structure factors of the reported AtNBR1–PB1 X-ray crystal structure is PDB ID 6TGS. Raw X-ray diffraction images of the AtNBR1–PB1 crystal structure (PDB ID 6TGS [https://doi.org/10.2210/pdb6tgs/pdb]) have been deposited under https://doi.org/10.5281/zenodo.3556558. The EMDB accession numbers for the L- and S-type AtNBR1–PB1 cryo-EM maps and models are EMD-10499/EMD-10500 and the corresponding PDB IDs 6TGN [https://doi.org/10.2210/pdb6tgn/pdb]/6TGP [https://doi.org/10.2210/pdb6tgp/pdb]. For the L- and S-type p62-PB1 cryo-EM maps and atomic coordinate models EMD-10501/EMD-10502 and PDB IDs 6TGY [https://doi.org/10.2210/pdb6tgy/pdb]/6TH3 [https://doi.org/10.2210/pdb6th3/pdb] have been assigned, respectively. The source data underlying Figs. 1C, 3F, 4C, 4D, 4F, 5A, 5E, and Supplementary Figs. 5F, 7D, 7E, and 7F are provided as a Source Data file. All relevant data are available from the corresponding author upon reasonable request.

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

## Acknowledgements

We thank the European Synchrotron Radiation Facility (ESRF, Grenoble, France), the EMBL beamlines at PETRA-III (DESY, Hamburg, Germany), and the beamline scientists at ESRF ID23-2 and EMBL-DESY P14 for excellent support. The project was financially supported by the Boehringer Ingelheim Fund's Exploration Grant. A.J.J. acknowledges financial support by an EMBL Interdisciplinary Postdoc (EIPOD) fellowship under Marie Curie Actions (PCOFUND-GA-2008-229597), a Marie Sklodowska-Curie IEF

fellowship (PIEF-GA-2012-331285), the Deutsche Forschungsgemeinschaft (DFG) through the excellence cluster "The Hamburg Center for Ultrafast Imaging (CUI)— Structure, Dynamics and Control of Matter at the Atomic Scale" (EXC1074), and the Joachim Herz Foundation. Work in the lab of T.J. was funded by grants from the FRIBIOMED (grant number 214448) and the TOPPFORSK (grant number 249884) programs of the Research Council of Norway to T.J.

## Author contributions

A.J.J. and C.S. designed research. A.J.J., T.K. and S.M. purified proteins and performed biochemical/biophysical characterization. A.J.J., W.J.H.H. and S.T.H. determined the cryo-EM structures. A.J.J. and M.W. determined the X-ray crystal structure. S.W.B. and A.B. performed and evaluated cellular EM. A.P., B.K.S., T.L. and T.J. designed and carried out experiments using cellular light microscopy. A.J.J. and C.S. wrote the paper with major input from all other authors.

## Competing interests

The authors declare no competing interests.
