## [Peer Review File · Nature Communications]

Reviewers' comments:

Reviewer #1 (Remarks to the Author):

In the present manuscript, Jakobi and colleagues provide an update on their previous observations about p62 filaments (Ciuffa et al., 2015, Cell Reports). This work provides strong structural data on the filament formation of p62-PB1 domain, as well as other similar PB1 domains. The authors also show that the presence of interactors and changes on the domain architecture of p62 affect the filament formation. However, besides fine structures of PB1 domains of p62 and AtNbr1, the biological analyses are too preliminary to support the author's hypothesis on the role of the PB1 domain and filament formation in the functions of p62. The manuscript, thus, is not acceptable in its current form for its publication in Nature Communications.

Some points that should be strengthened before considering a resubmission would be:

1. The authors showed the importance of double R finger in the polymer formation of the PB1 domains on the basis of their fine structural analyses, but they failed to show its importance in autophagy and the liquid-liquid phase separation in cells.
2. The authors examined the level of Keap1 in p62-deficient HeLa cells expressing wild-type p62 or a series of p62 mutants including the PB1 chimera, but no quantification is provided regarding the size and number of p62 bodies in each case.
3. The p62 protein is itself a selective substrate for autophagy. The authors should investigate the degradation of wild-type and mutant p62 as well as their structures in cells. Pulse-chase experiments by western blot could be a valid approach to measure the degradation rate of each of the p62 constructs assayed.
4. As the authors have shown in previous reports, p62 serves as a receptor for ubiquitinated cargos and is a multivalent protein that has an ability to form liquid-droplets. The authors should examine whether the p62 mutants including the PB1 chimera mutants have an effect on autophagic degradation of ubiquitinated proteins and on the liquid-droplets. To strengthen the link between filaments and p62 bodies, the authors should provide evidence that the filament-containing structures they observe have liquid-like properties (measured by FRAP assay, in vivo fusion-fission of the bodies...).

Reviewer #2 (Remarks to the Author):

Jacobi et al describe in their manuscript how the autophagy receptor p62/SQSTM1 can induce formation of phase-separated p62 bodies by using its PB1 domain in vitro. They show by AUC and negative stain EM that the PB1 domain of Arabidopsis AtNBR1 can form filamentous tubular assemblies. Then they go on to determine the structures of these filaments in 4 forms (L and S, human and plant) by cryo-EM, and identify key interactions. Using ITC and negative stain EM, they characterize inter p62-PB1 domain interactions and find preferred binding to one end. Going beyond this, they used correlative LM-EM (CLEM) to study the ultrastructure of phase-separated p62 bodies in vitro by electron tomography. For the first time, they directly observe the filamentous structure of these p62 bodies by cryo-ET and quantitate the length of their filaments. Furthermore, they study the relevance of the filamentous assembly states within the p62 bodies in cells for lysosomal targeting in the autophagy pathway by creating fusion chimeras between p62 and TFG1, resulting in very different filament types, which they subsequently show by confocal LM to be able to form p62 bodies in cells. Finally, they demonstrate that this chimera can be turned over by autophagy using fluorescence microscopy, but that it cannot act as a cargo receptor for the p62-specific substrate KEAP1 to mediate autophagy degradation, further establishing the filamentous state of p62 as required for autophagosomal processing.

The paper is very well written and follows thorough scientific methodology by implementing systematic controls, as well as sequentially building upon previous results by others and the present work itself. The amount of techniques and modalities used in the study is breathtaking – each of them requires in-depth expertise and may have been published by other authors as separate papers. Nevertheless, they were executed in a flawless synergistic manner and result in the compelling story presented in this work. This work is a shining example how cryo-EM can be used beyond mere structure determination to enable insights into fundamental biological questions. Autophagy is an emerging field and is increasingly recognized as highly important and

impactful. I have thoroughly enjoyed reading this pioneering article despite its intricate topic, whose strings have been clearly presented and entangled by the authors. In fact, I have not found a flaw in this impressive study and I recommend to publish it as is without further review.

A few formal or minor comments:

Line 148: PB1polymerize, insert space

Line 578: focussed, spelling

Lines 285-288: may be useful to add one additional sentence explaining why TGF1 was chosen.

Fig. 6C: scale bar

Reviewer #3 (Remarks to the Author):

The manuscript "Structural basis of p62/SQSTM1 helical filaments, their presence in p62 bodies and role in cargo recognition in the cell" describes structural studies, using cryo-EM, of the PB1 domain of p62/SQSTM1 in vitro and in cells. PB1 domains of p62 and other PB1-domain containing proteins have been subjected to cryo-EM in vitro and filaments of varying sizes have been observed for the various domains. In a next step, the authors try to recapitulate the differences of these filamentous structures in cells and test the hypothesis whether the filament structure predicts p62 function by generating hybrid chimeras of p62 with PB1 domains of various structures. Generally, the paper is well done and interesting.

My main concern is that the conclusion "The precise structural context within the filament assembly is required for specific cargo recognition of KEAP1" is not fully supported by the experiments. This conclusion is very tempting, but it is correlative in nature and there may be other factors involved such as the sequence of the swapped PB1 domains.

Some minor points:

1. KEAP1 should be introduced in the Introduction;
2. Five PB1 domains were purified, but only three are shown in Figure 1. I take it that the others don't form filamentous or tubular structures. Is that also true for p62 (1-102). It might be worth pointing this out in more detail and possibly discuss why this might be the case.
3. Which microscope was used in Figure 6C? The scale bars are missing. If it is a conventional confocal microscope, I would think that the resolution is not able to resolve structure of 0.1-0.5 μ m size. Also, the legend does not comment on the size of the scale bars in Figure 6F,G.

We would like to thank the Referees for their overall positive feedback on the manuscript and have taken their concerns very seriously. In the course of the revision process, we included two completely new Supplementary Figures, two updated panels and two Movies based on new cellular imaging and biochemistry data. We give a detailed point-by-point response below:

Response to Referee 1:

1. The authors showed the importance of double R finger in the polymer formation of the PB1 domains on the basis of their fine structural analyses, but they failed to show its importance in autophagy and the liquid-liquid phase separation in cells.

We agree with the reviewer that these are essential questions, and for the revision we have performed cellular experiments with the R21A/R22A mutant of p62 in order to assess its cellular behavior in autophagy and in liquid-liquid phase separation. We found that the R21A/R22A mutant (GFP-p62 R21A/R22A) was completely diffuse when expressed in cells (New Supplementary Figures 7 and 8), indicating that the RR finger is indeed essential for liquid-liquid phase separation in cells. Furthermore, the double tag construct (mCherry-GFP-p62 R21A/R22A) did not accumulate in red only dots (New Supplementary Figure 7B), indicating that the RR finger is essential for degradation of p62 by autophagy. Since the R21A/R22A mutant was completely diffuse when expressed in cells, we have not included data for this construct in our bar diagrams displaying puncta formation (there was no puncta to count). As expected, the R21A/R22A mutant also failed to promote aggregation of KEAP1 or autophagic degradation of KEAP1 in co-transfected cells (New Supplementary Figure 7C).

For the revised manuscript, we also made HeLa FlpIn T-Rex p62 KO cells expressing GFP-p62 or GFP-p62 R21A/R22A from a tetracycline induced promoter. Our data from live cell imaging and flow cytometry analyses of these cells (New Supplementary Figure 8 and Movies 1 and 2) confirmed that the RR finger is essential both for the aggregation of p62 in p62 bodies and for the degradation of p62 by autophagy.

2. The authors examined the level of Keap1 in p62-deficient HeLa cells expressing wild-type p62 or a series of p62 mutants including the PB1 chimera, but no quantification is provided regarding the size and number of p62 bodies in each case. <

In our original manuscript, we analyzed the aggregation of the different p62 and TFG1-p62 constructs (Figures 6C-D and Supplementary Figure 5F). In our revised manuscript, we have included information on the average number of dots per cell in our revised Supplementary Figure 5A (not included in our original manuscript). We have also improved the text lines in Figures 6D, 6E and 6H, so that the bar diagrams are more easily understood.

In our original manuscript, we did not similarly analyze the size and number of p62 bodies in cells co-expressing KEAP1. Importantly, when we analyzed the aggregates, we observed that for all constructs the co-localization with KEAP1 was 100%. All aggregates contained both proteins, and when analyzing single aggregates, the co-localization was 100% within each aggregate (see representative images in Supplementary Figure 6). Hence, an analysis of KEAP1 puncta is an analysis of p62-KEAP1 puncta. We also observed that the co-expression of KEAP1 increased the

size of p62 and TFG1-p62 aggregates, but the effect was similar for all constructs. KEAP1 is part of an E3 ligase complex known to ubiquitinate p62 bodies (Lee et al., 2017; Cell Rep., 19:188-202). The presence of KEAP1 will therefore affect ubiquitination of p62 and TFG1-p62 aggregates, and this presumably explains why the size of aggregates increases in cells co-expressing KEAP1. Since we observed no notable difference in how the different constructs aggregated in the presence of KEAP1, we do not show data on the size and number of p62 bodies in cells co-expressing KEAP1. The main conclusion is that all constructs co-aggregated efficiently with KEAP1.

3. The p62 protein is itself a selective substrate for autophagy. The authors should investigate the degradation of wild-type and mutant p62 as well as their structures in cells. Pulse-chase experiments by western blot could be a valid approach to measure the degradation rate of each of the p62 constructs assayed.

The proposed pulse-chase experiments will illuminate the degradation rates of the p62 constructs. As we did not investigate or speculate on any time aspects of degradation in the manuscript, we feel that such an experiment will not be able to add or consolidate our major conclusions of the manuscript. Although it is clearly an interesting aspect to further investigate, it is beyond the scope of this manuscript.

4. As the authors have shown in previous reports, p62 serves as a receptor for ubiquitinated cargos and is a multivalent protein that has an ability to form liquid-droplets. The authors should examine whether the p62 mutants including the PB1 chimera mutants have an effect on autophagic degradation of ubiquitinated proteins and on the liquid-droplets. To strengthen the link between filaments and p62 bodies, the authors should provide evidence that the filament-containing structures they observe have liquid-like properties (measured by FRAP assay, in vivo fusion-fission of the bodies...).

After the in vitro demonstration of liquid-like properties of p62 filaments by us and others (Zaffagnini et al., 2018 and Li et al., 2018), it would indeed be interesting to characterize the exchange properties of the EM-imaged p62 bodies. Unfortunately, to our knowledge there is currently no correlative technique that can combine live-cell imaging with cellular electron microscopy as the samples need to be fixed. Therefore, we feel this request goes beyond the scope of the manuscript.

Response to Referee 2:

A few formal or minor comments:

Line 148: PB1polymerize, insert space.

This is now corrected.

Line 578: focussed, spelling.

This is now corrected.

Lines 285-288: may be useful to add one additional sentence explaining why TGF1 was chosen:

As suggested, we added the following statement to motivate more clearly:

Therefore, we reasoned that a p62 chimera, in which we exchange the native PB1 domain for TFG1-PB1, could clarify the role of the helical PB1 scaffold in autophagy clearance.

Fig. 6C: scale bar

This is now corrected.

Response to Referee 3:

My main concern is that the conclusion “The precise structural context within the filament assembly is required for specific cargo recognition of KEAP1” is not fully supported by the experiments. This conclusion is very tempting, but it is correlative in nature and there may be other factors involved such as the sequence of the swapped PB1 domains.

We thank the Referee for pointing out this. We agree that it is probably not the cargo recognition as such which is detrimentally affected here but rather the cargo uptake and degradation. As we do observe efficient recruitment of KEAP1 into aggregates formed by the chimera constructs, our data do not support any defect in the ability to aggregate with KEAP1. In addition, we now verified experimentally that the purified KEAP1-DC domain binds to the TFG1-chimera as well as to p62 WT using pull-down experiment (*New Supplementary Figure 5G*). Hence, in the revised manuscript we now state this conclusion more carefully by amending the abstract and also by replacing the sentence in question more carefully: “Also, the precise structural context of the filament assembly is affecting the ability to degrade KEAP1-containing aggregates, and possibly other p62-specific cargoes.”

Some minor points:

1. KEAP1 should be introduced in the Introduction;

We agree, and this is now done in the revised manuscript.

2. Five PB1 domains were purified, but only three are shown in Figure 1. I take it that the others don't form filamentous or tubular structures. Is that also true for p62 (1-102). It might be worth pointing this out in more detail and possibly discuss why this might be the case.

Indeed, this question could come up in the comparison of the different PB1 domains. In our previous study (Ciuffa et al. 2015), we showed that both p62(1-102) and p62(1-122) formed tubular structures. Therefore, we added a clarifying statement after the pelleting assay to the manuscript. The here-presented p62 (1-122) micrograph shows higher homogeneity and therefore we here preferred to work with the slightly longer p62 construct.

3. Which microscope was used in Figure 6C? The scale bars are missing. If it is a conventional confocal microscope, I would think that the resolution is not able to resolve structure of 0.1-0.5 μm size. Also, the legend does not comment on the size of the scale bars in Figure 6F,G.

Corrected. All scale bars in figure 6C, 6F and 6G are 10 μm , as are the scale bars in Supplementary Figures 5 and 6. We have used Zeiss 780 and magnifications 63x oil. Seeing the length of the scale bar, it is no doubt that we can observe dots smaller than 0.5 μm , but we agree that 0.1 μm is difficult to see. We have changed the text in the x axis in figure 6C from 0.1-0.5 to <0.5.

REVIEWERS' COMMENTS:

Reviewer #1 (Remarks to the Author):

Although they did not do all the experiments I requested, the manuscript reaches to be acceptable for publication in Nature Communications.

Reviewer #3 (Remarks to the Author):

I am satisfied with the revisions.